# A Bioactive Substance Derived from Brown Seaweeds: Phlorotannins

**DOI:** 10.3390/md20120742

**Published:** 2022-11-26

**Authors:** Hongli Zheng, Yanan Zhao, Lei Guo

**Affiliations:** 1Jiangsu Key Laboratory of Marine Bioresources and Environment, Co-Innovation Center of Jiangsu Marine Bio-Industry Technology, Jiangsu Ocean University, Lianyungang 222005, China; 2Jiangsu Key Laboratory of Marine Biotechnology, School of Food Science and Engineering, Jiangsu Ocean University, Lianyungang 222005, China

**Keywords:** phlorotannins, extraction technology, separation and purification, biological activity

## Abstract

Phlorotannins are a type of natural active substance extracted from brown algae, which belong to a type of important plant polyphenol. Phloroglucinol is the basic unit in its structure. Phlorotannins have a wide range of biological activities, such as antioxidant, antibacterial, antiviral, anti-tumor, anti-hypertensive, hypoglycemic, whitening, anti-allergic and anti-inflammatory, etc. Phlorotannins are mainly used in the fields of medicine, food and cosmetics. This paper reviews the research progress of extraction, separation technology and biological activity of phlorotannins, which will help the scientific community investigate the greater biological significance of phlorotannins.

## 1. Introduction

Brown seaweeds are a kind of economic algae widely distributed in cold water areas of the sea and have great potential for development. Common brown seaweeds are *Laminaria japonica*, *Undaria pinnatifida*, *Pelvetia siliquosa*, *Hizikia fusifarme*, *Ecklonia kurome*, *Sargassum thunbergii*, *Sargassum horneri*, *Chorda filum*, *Scytosiphon lomentarius*, *Sargassum miyabei* and so on [1,2,3,4]. Brown algae are rich in seaweed polysaccharides, proteins, amino acids, polyphenols, terpenes, mannitol, fucoxanthin, hormones and other active substances. Among them, the polyphenolic compounds have the characteristics of special structure, rich biological activity and high medical value.

Brown algae polyphenols are a class of polymers with phloroglucinol as the basic unit, which are widely distributed in brown algae and are an important component of brown algae cell-walls [5,6]. Brown algae polyphenols have many activities, including anti-oxidation, anti-tumor and anti-cancer, improving cardiovascular function, antibacterial, anti-virus, chemical defense, deodorization, anti-fatigue, anti-obesity, anti-diabetes and immune enhancement [7]. Many brown algae contain unique brown algae polyphenols known as “phlorotannins”. The European Union has approved phlorotannins as a new resource food and they are allowed to be used in dietary supplements. Brown algae polyphenols have great potential to become new medicines and health products due to their rich biological activities and their widespread characteristics in nature.

According to research, brown algae polyphenols can be divided into two types, those of low molecular weight and those of high molecular weight. In addition to phloroglucinol (structural unit), the structures of low molecular-weight brown algae polyphenols can be divided into the following categories (Figure 1 and Figure 2) according to their polymerization methods [8].

Oligomeric phloroglucinol can be further polymerized to form tannins with a higher molecular weight, which are basically the same as terrestrial tannins. At present, the molecular structure of brown algae polyphenols has not been clearly confirmed by analytical techniques such as infrared, ultraviolet and nuclear magnetic resonance techniques. This paper briefly reviews the research progress of the extraction process, separation, purification technology and biological activity of brown algae polyphenols at home and abroad in recent years, in order to provide a reference for the development and research of marine brown algae resources.

## 2. Extraction Process

### 2.1. Solid–Liquid Extraction Method

Solid–liquid extraction (SLE), also known as organic solvent extraction, is one of the main methods for extracting active ingredients from plants. Most terrestrial Chinese herbal medicines and algae use this method to obtain active substances. The biggest advantage of this method is that the rationale of the method is clear, the operation is simple, the steps are fixed, and the requirements for instruments and equipment are low. The key to organic solvent extraction is the selection of organic solvents and the screening of extraction conditions (temperature, time, sample pretreatment, etc.), which directly affect the extraction efficiency and purity of active substances. Among them, the organic solvent commonly used in brown algae extraction is ethanol. Recent advances in the use of SLE to extract marine phenolics from seaweeds are shown in Table 1.

Different solvents and temperatures had significant effects on the content of the extracted phenolic compounds. Comparing the acidic solvent with pure water, it was found that the content of phenols extracted by the acidic solvent was lower. This indicates that acidic solvents at high temperature (70 °C) may affect the extraction of phenolic compounds, and the combination of acidic media with a high temperature may lead to the hydrolysis of complex phenolic structures into simpler ones [9]. The higher the temperature, the higher the yield of phenolic substances in water, and the higher the yield of ethanol extract [10]. Moreover, there is a type of promising green solvent for extracting biologically active compounds from plant raw materials, which are natural deep eutectic solvents (NADES). The efficiency of extracting active substances depends on the polarity of NADES, which is closely related to their solubilizing capacity. The polarity of NADES is more suitable for extracting phenolic compounds from plants than H_2_O, and its extraction efficiency is better. The mole ratios of components in the NADES could also considerably affect their extraction capacity. Aqueous solutions of NADES based on choline chloride and lactic acid (1:1~1:3) were most efficient for extracting phlorotannins from the studied brown algae species [11]. Lactic acid: choline chloride and lactic acid: glucose: water-based NADES are suitable for the simultaneous extraction of hydrophilic (ascorbic acid and phlorotannins) and lipophilic (fucoxanthin) compounds from *Fucus vesiculosus*. In addition to reducing the impact on the environment, NADES enables the high stability of the active metabolite of *F*. *vesiculosus* and retains the antioxidant activity of the extract [12].

**Table 1 marinedrugs-20-00742-t001:** Solid–liquid extraction (SLE) of marine phenolics.

Seaweed Type	State of the Seaweed (Wet/Dry/Particle Size)	Organic Solvents Used	Solvent–Seaweed Ratio/Temperature (°C)/Time	Yield	Application of the Extract	Reference
*Undaria* *pinnatifida*	Air-dried at 25 °C and ground to fine powder (50 µm)	Ice-cold n-hexane	15:1/on an ice bath/20 min (three times)	0.5%	Antioxidant and anti-inflammatory activity	[1]
*Fucus* *vesiculosus*	Freeze-dried, ground to a powder and stored in vacuum-packed bags at −80 °C prior to extraction	Ethanol/water (80:20)	10:1/room temperature/24 h	231.95 ± 8.97 μg PGE/mg sample (PGE means phloroglucinol equivalent)	Antioxidant activity	[5]
*Fucus* *vesiculosus*, *Ascophyllum nodosum*	Powder	Aqueous NADES (natural deep eutectic solvents) solutions (50–70%)	5:1/50 °C/2 h	60–72%	/	[11]
*Sargassum* *fusiforme*	Particles lower than 0.8 mm; stored under vacuum (75%) at −20 °C	30% ethanol-water mixture	5:1/25 °C/30 min	63.35 ± 0.19 mg PGE·g^−1^	Antioxidant activity	[13]
*Fucus* *vesiculosus*	Freeze dried/powdered	Acetone 67% (*v*/*v*)	70 mL/g/25 °C/3 h	2.92 ± 0.05 mg GAE/g DW (GAE means gallic acid equivalent; DW means dry weight)	The adjustment of activities of α-glucosidase, α-amylase and pancreatic lipase	[14]
*Pelvetia canaliculate*	Freeze-dried macroalgal material	Ethanol-water (80:20) (EW)	10:1/room temperature/24 h	275.13 ± 0.41 mg PGE/g DW	Antioxidant activities	[15]
*Fucus spiralis*				278.81 ± 0.61 mg PGE/g DW		
*Ascophyllum nodosum*				278.77 ± 0.26 mg PGE/g DW		
*Carpophyllum flexuosum*	Oven-dried biomass	Acidified methanol (HPLC grade, 50%, aqueous solution; pH 2), followed by acetone (AR grade, 70%, aqueous solution; pH 2)	40:1/room temperature/24 h, respectively	8.6 ± 0.2% mg PGE/g DW	Antioxidant activities	[16]
*Carpophyllum plumosum*				7.5 ± 0.1% mg PGE/g DW		
*Eisenia bicyclis*	Fresh sample was washed with fresh water for 3 h by soaking	100% ethanol	2:1/room temperature/6 h	/	/	[17]
*Carpophyllum flexuosum*	Powder through a 1.0 mm sieve stored at −20 °C in sealed bags Powder through a 1.0 mm sieve stored at −20 °C in sealed bags	Acetone/water (7:3, acidified to pH 2 with 2 M HCl)	30:1/room temperature/24 h	8.6 ± 0.2 (PGE % of DW)	Antioxidant activities	[18]
*Carpophyllum plumosum*				7.5 ± 0.1 (PGE % of DW)		
*Ecklonia radiata*				1.5 ± 0.2 (PGE % of DW)		

### 2.2. Ultrasonic-Assisted Extraction

Ultrasonic-assisted extraction (UAE) mainly relies on the cavitation effect, mechanical effect and thermal effect of ultrasonic waves to rapidly release intracellular substances, diffuse and dissolve in organic solvents, thereby significantly improving the extraction efficiency of active substances [19]. The advantages of this method lie in short time required, high extraction rate and low heating temperature, which can protect substances that are easily oxidized, decomposed or hydrolyzed, and thermally unstable substances from being damaged during the extraction process. Some studies that make use of UAE to extract marine polyphenols from seaweed are shown in Table 2.

UAE was found to be more effective than conventional solvent for extraction of bioactive substances. Different process parameters that affect extraction include temperature, ultrasound frequency and duration of extraction, and the performance of the extraction solvent (such as viscosity and surface tension) [20].

**Table 2 marinedrugs-20-00742-t002:** Ultrasound-assisted extraction (UAE) of marine phenolics.

Seaweed Type	State of the Seaweed (Wet/Dry/Particle Size)	Organic Solvents Used	Solvent–Seaweed Ratio/Temperature (°C)/Time (min)/Power(w)	Yield (mg GAE/g DW)	Application of the Extract	Reference
*Bifurcaria bifurcata*	Particles lower than 0.8 mm; stored under vacuum (75%) at −20 °C	Water/ethanol (50:50, *v*/*v*)	10:1/-/30 min/-	5.74 g (GAE/100 g dried seaweed)	Antioxidant activity	[21]
*Fucus vesiculosus*	Dried at 50 °C, 9 days, and milled to 1 mm particle size; stored at room temperature in dark conditions	50% ethanol	10:1/30 min//35 kHz	TPC: 572.3 ± 3.2 mg GAE/g (TPC means total phenolic compound)	Antioxidant activity	[22]
*Silvetia compresssa*	Samples were ground, sieved with a 500-μm sieve, and vacuum packed until use	32.33% aqueous ethanol	30 mL·g^−1^/30 min/ice bath /3.8 W·cL^−1^	/	Potential prebiotic effect	[23]
*Hormosira banksii*	The dried sample was ground to give ≤0.6-mm particle size	Ethanol 70% (*v*/*v*)	50 (mL·g^−1^)/60 min/30°C/60% or 150 W	24.07 mg GAE/g	Antioxidant activity	[24]
*Ascophyllum nodosum*	Powder through a 0.5 mm mesh.	Acid concentration (0.03 M HCl)	10:1/25 min/ultrasonic amplitude (114 μm)	143.12 mg GAE/g	/	[25]
*Ascophyllum nodosum*	/	Water and water + HCl (0.03 M)	-/10 min/-/20 kHz/750 W	82.70 mg GAE/g	/	[26]
*Ecklonia arborea*	Powdered	50% ethanol	20:1/30 min/70 °C/-	179.16 ± 11.38 (GAE)/L	Antiviral activity	[27]
*Silvetia filiformis*				102.22 ± 15.10 (GAE)/L		
*Ecklonia cava*	Powdered	80% MeOH	20:1/2 days/room temperature/-	/	Antiviral activities against H1N1 and H9N2	[28]
*Ecklonia maxima*	Pieces	Ethanol: water (80:20)	10:3/1.5 h/25 °C/-	/	Neuroprotective agent	[29]

### 2.3. Microwave-Assisted Extraction

Microwave-assisted extraction (MAE) mainly uses the action of an electromagnetic field to effectively separate the active ingredient from the matrix. It is a separation and extraction method that does not destroy the original state of the extracted substance. Compared with the traditional solvent extraction method, this method has the advantages of uniform heating of the system, high extraction rate, short extraction time, good stability and good selectivity. When compared with ultrasonic extraction, it is found that microwave assistance takes less time and is more efficient [5]. It is suitable for the extraction of heat-sensitive substances and thermally unstable substances, such as phenols and natural pigments. A few of the studies with regard to the extraction of marine phenols in use of MAE can be seen in Table 3.

MAE belongs to selective internal heating, with short extraction time, simple operation, easy to control, continuous extraction characteristics, especially suitable for fresh plant extraction, because water can effectively absorb the energy of microwave radiation, more suitable for industrial continuous production. Studies have shown that the combination of continuous reflux extraction and microwave radiation can significantly shorten the extraction time and greatly reduce environmental pollution [30].

### 2.4. Accelerated Solvent Extraction

Accelerated solvent extraction (ASE), also known as pressurized liquid extraction (PLE), is a short-time, low-solvent consumption and high-efficiency method in the extraction of phenolic compounds. The technology uses solvents at high pressure (1000 to 3000 PSI) and temperature (50 to 200 °C), which speeds up the extraction process. The increased temperature can greatly attenuate the interaction forces caused by van der Waals forces, hydrogen bonds, target molecules and dipole attraction at the active location of the sample matrix. The solubility of the liquid is much greater than the solubility of the gas, so increasing the pressure in the extraction tank makes the solvent temperature higher than the boiling point under its atmospheric pressure. The advantages of this method are the low amount of organic solvent required, rapidity, low matrix influence, high recovery rate and good reproducibility. Some of the studies that demonstrate the extraction of marine polyphenols by the ASE method are shown in Table 4.

The high temperature used increases solubility while reducing solvent viscosity, enhancing sample diffusion and penetration, and facilitating desorption of target components. Despite the use of high temperature, in ASE, the degradation of phenolic compounds does not occur due to the absence of air and light, which is one of the main advantages of this technique [35].

**Table 4 marinedrugs-20-00742-t004:** Accelerated solvent extraction (ASE) of marine phenolics.

Seaweed Type	State of the Seaweed (Wet/Dry/ Particle Size)	Organic Solvents Used	Solvent–Seaweed Ratio/Temperature(°C)/Time (min)/Pressure	Yield	Application of the Extract	Reference
*Sargassum muticum*	Freeze-dried, reduced to powder and finally sieved at 250-μm	95% ethanol	11:2/160 °C/20 min/1500 psi	94.0 mg GAE/g DW	/	[10]
*Undaria pinnatifida*	Dried in an oven at 80 °C for 48 h (through a 40-mesh sieve)	52% ethanol	10:1/170 °C/5.2 h	10.7 ± 0.2 mg GAE/g DW	Anti-inflammatory and antioxidant activity	[36]
*Ascophyllum nodosum*	Powder	Ethanol	/	50.2 mg GAE/g	Antibacterial activity was tested against food spoilage bacteria	[37]
*Sargassum muticum*	Freeze-dried, reduced to powder and finally sieved at 250-μm	75% ethanol	11:2/120 °C/20 min/1500 psi	47.55 ± 2.28 mg GAE/g DW	/	[38]
*Laminaria ochroleuca*	Powder	50% ethanol	-/80,120,160 °C/10 min/1500 psi	173.65 mg GAE/g PLE extract	/	[39]
*Fucus vesiculosus*	Freeze-dried and ground to a semi-coarse powder (through a 1.0 mm sieve)	Dichloromethane and methanol	20:1/-/5 + 5 min/1500 psi	-	Methicillin-resistant *Staphylococcus aureus* (MRSA) growth inhibitory, radical scavenging, and pro-apoptotic activities	[40]

### 2.5. Supercritical Fluid Extraction

Supercritical fluid extraction (SFE) is an extraction technology that replaces organic solvents with supercritical fluids to extract active ingredients from solid or liquid substances. The supercritical fluid is a gas at room temperature and normal pressure, and is easily separated from the extracted residual phase and extracted components after extraction. The method has the characteristics of a simple operation process, good extraction effect, no solvent residue, etc. It also can avoid the concentration process of removing most of the extraction solvent. Modern supercritical fluid extraction technology usually uses supercritical CO_2_ as an extractant because of its favorable critical temperature (304.1 K) and pressure (73.8 bar) that enable the active ingredients to be extracted from thermally unstable materials such as biomolecules [41]. The relevant studies which use SFE to obtain marine polyphenols are shown in Table 5.

The study also showed that supercritical CO_2_ (SC-CO_2_) extraction is not suitable for the recovery of phlorotannins from brown algae. However, the addition of water and ethanol as co-solvents in SC-CO_2_ extraction has been shown to significantly increase the polyphenol yields obtained with pure CO_2_ [42,43].

### 2.6. Biological Enzyme Extraction

Biological enzyme extraction (EAE) is a type of green extraction technology. In plants and algae, the corresponding hydrolase (such as cellulase) is selected to dissolve the cell wall structure of the raw material, so that more plant cell contents are dissolved or miscible in organic solvents, so as to achieve the purpose of separating active ingredients and improving the extraction efficiency [45]. The method releases the active substances in the plant well and improves the extraction efficiency of the compounds. At the same time, it can also decompose macromolecular substances by enzymatic hydrolysis, so that the active ingredients and functional factors tightly wrapped by colloid can be released, which is convenient for dissolution and extraction. Some studies using EAE to obtain marine polyphenols are shown in Table 6.

The biggest feature of EAE is the high extraction rate and high yield, which makes it an economical and sustainable method. However, the selection of carbohydrate enzymes and the optimization of hydrolysis conditions, such as enzyme–substrate ratio, temperature, hydrolysis time, stirring speed and pH, are critical to maximizing the yield of bioactive compound recovery [46].

**Table 6 marinedrugs-20-00742-t006:** Enzyme-assisted extraction (EAE) of marine phenolics.

Seaweed Type	State of the Seaweed (Wet/Dry/Particle Size)	Type of Enzyme Used	Extraction Conditions Enzyme Conc./Temperature/Time/pH	Yield	Application of the Extract	Reference
*Ulva armoricana*	Powder	Neutral endo-protease	6% (*w*/*w*, DW)/50 °C/240 min/pH 6.2	9 mg GAE/g DW	Antioxidant and antiviral activities	[47]
		A mix of neutral and alkaline endo-proteases		11 mg GAE/g DW		
		A multiple-mix of carbohydrase		7 mg GAE/g DW		
		Mix of endo-1,4-β-xylanase/endo-1,3(4)-β-glucanase		6 mg GAE/g DW		
		Cellulase		4 mg GAE/g DW		
		Exo-β-1,3(4)-glucanase		7 mg GAE/g DW		
*Sargassum angustifolium*	Freeze dried/ powdered	Viscozyme^®^	0.1%/50 °C/20 h/pH 4.5 (0.1 M acetate buffer)	8250 mg GAE/100 g DW	Antioxidant and antimicrobial activities	[48]
*Juvenile Lessonia nigrescens*	Powder	α-Amylase	1:10 (S/L)/60 °C/17 h/pH 6 and in phosphate buffer (0.2 N)	37.72 ± 4.13% of the dry weight (DW)	ACE inhibitory activity	[49]
*Macrocystis pyrifera,*		Cellulose	1:10 (S/L)/50 °C/17 h/pH 4.5 and in acetate buffer (0.1 N)	35.36 ± 0.54% DW		
*Durvillaea antarctica*		α-Amylase	1:10 (S/L)/60 °C/17 h/pH 6 and in phosphate buffer (0.2 N)	9.07 ± 0.78% DW		
*Macrocystis* *pyrifera*	Dried at room temperature, and powdered (<0.5 mm)	Alginate lyase, fucoidanase, and 1,3-β-D-glucanase	1:20 (S/L)/25 °C/36 h/pH 7.0	2.14 ± 0.25 wt%	Radical scavenging activity, total antioxidant activity (TAA)	[50]
*Hizikia fusiformis*	/	Endo-peptidase and β-glucanase	-/50–60 °C/24 h/7.0–8.0	/	Antioxidant activity	[51]
*Codium tomentosum*	/	Cellulase and viscozyme^®^	48–62%/50 °C/24 h ± 60 min (pauses of 2 min after each 10 min of sonication)/400 w ultrasonic processing/-	261 ± 37 μg catechol equip/g lyophilized extract	Inhibitory potential against α-glucosidase	[52]

## 3. Separation and Purification Technology

### 3.1. Silica Gel Column Chromatography

Silica gel column chromatography, also known as silica gel adsorption chromatography technology, is a separation technology based on the principle of distribution equilibrium. The adsorption is mainly due to the van der Waals forces and hydrogen bonding between silica gel and the components to be separated. The silica gel column chromatography method is mainly divided into the normal phase method and the reversed phase method; additionally, the stationary phase commonly used in the normal phase method is silica gel, alumina and polyamide adsorption column. This method is suitable for polyphenol compounds with large polarity, and the separation effect is remarkable. Repeated gradient elution or isobaric elution can separate polyphenolic compounds of different structures. The reversed phase method is suitable for substances that are difficult to separate by the normal phase method, and the commonly used eluents are methanol–water systems of different proportions, and the polar substances are first eluted. Active ethyl acetate (EtOAc) fraction (51.5 g) obtained from *Eisenia bicyclis* was subjected to chromatography on a silica gel column, with EtOAc–MeOH (15:1) as the eluent, yielding fractions EF08 (3.98 g). After repeat column chromatography of EF08 with a solvent mixture of EtOAc and MeOH yields five sub-fractions (EF0801-EF0805). Among them, EF0804 (0.41 g) was purified on an RP-18 column and eluted with aqueous MeOH (20% MeOH–100% MeOH, gradient elution) to yield a phlorotannin, 974-B [53].

### 3.2. Macroporous Adsorption Resin

Macroporous resin, also known as large grid adsorbent, is a separation material that combines adsorption and screening. Macroporous resin is a nonionic polymer with stable properties. It is insoluble in alkalis and organic solvents, nor does it react with inorganic salts, strongly ionic compounds or low molecular weight compounds. Now it is widely used in the separation, extraction and purification of natural products. To purify the phlorotannins extracted from *Macrocystis*, Leyton et al. [54] evaluated the adsorption capacity of six resins (HP-20, SP-850, XAD-7, XAD-16N, XAD-4, and XAD-2). Under the same adsorption time, XAD-16N-purified phlorotannins had the highest efficiency of 42%. The macroporous adsorption resin HP-20 as an adsorbent and methanol–water system as an elution agent could isolate the active ingredients in *Macrocystis pyrifera*.

### 3.3. Thin Layer Chromatography

Thin layer chromatography (TLC) makes use of the suitable stationary phase such as silica gel, diatomaceous earth, alumina, microcrystalline cellulose, etc., plus uniform adhesive grinding, and then coated on glass plates, plastic or aluminum substrates, with natural drying or drying to form a uniform thin layer. After the capillary spotting is unfolded in an organic solvent, the retardation factor (R_f_) is compared with the R_f_ of the reference substance solution, so as to identify the active substance, check the impurity and determine the content. This method is easy to operate, sensitive and fast. Thin layer chromatography can be combined with column chromatography [55] and the macroporous adsorption resin method [56] to improve the separation and purification effect of brown algae polyphenols.

### 3.4. Preparative High-Performance Liquid Chromatography

Preparative high-performance liquid chromatography (Prep-HPLC) is a liquid chromatography preparation method for purity separation on a large inner diameter, high load separation column, which can be highly extracted and purified for crude products. Its working principle is to take the liquid as the mobile phase, and use a high-pressure infusion pump to transform the mobile phase into the column. The composition and affinity of the mobile phase and fixed phase are different, so as to allow them to flow out of the stationary phase successively, to achieve the purpose of separation of each component [57]. With the octyldecylsilane (C18) column as the stationary phase, elution with methanol–water, 230 nm ultraviolet detection and the continuous optimization of the method, the isolation of monomeric phlorotannins can be obtained [58].

## 4. Chemical Characterization

### 4.1. Determination of Polyphenol Content of Brown Algae by Folin–Ciocalteu Method

The Folin–Ciocalteu method is a simple, rapid and inexpensive assay widely applied for the determination of total polyphenol content. Its principle of action is that phosphomolybdotungstic acid can quantitatively oxidize the polyphenolic compounds in an alkaline solution, and itself is reduced (turning W^6+^ into W^5+^) to form blue compounds [59]. The degree of blueness is proportional to the number of phenolic groups. The Folin–Ciocalteu assay relies on the transfer of electrons in an alkaline medium from phenolic compounds to phosphomolybdic/phosphotungstic acid complexes, which are determined spectroscopically at 765 nm. Although the electron transfer reaction is not specific for phenolic compounds, the extraction procedure eliminates approximately 85% of ascorbic acid and other potentially interfering compounds.

### 4.2. Determination of Polyphenol Content of Brown Algae by DMBA Method

The DMBA method [60] is determined by using 2,4-dimethoxybenzaldehyde (DMBA) to react specifically with 1, 3- and 1, 3, 5-substituted phlorotannins to form colored products. Measure the absorbance value at 520 nm. By plotting the standard curve between the concentration of the phloroglucinol standard and the absorbance of the reaction mixed solution, the brown algae polyphenol content calculated by the equivalent of the standard is obtained and the yield is calculated [61]. This new assay eliminates the problem of measurement interference, which is inexpensive, fast and can be used for small sample volumes.

### 4.3. High-Performance Liquid Chromatography Coupled with Mass Spectrometry

Identification and quantification of phlorotannins in brown algae are usually performed by reversed-phase high-performance liquid chromatography (RP-HPLC) in combination with methanol/acetonitrile and water (buffer) solvents and in the ultraviolet range of the spectrum [62]. HILIC columns utilize hydrophilic stationary-phase and reversed-phase eluents and can be described as variants of normal-phase HPLC that overlap with reversed-phase HPLC. Therefore, HILIC columns are ideal for isolating phlorotannins because they interact strongly with the polar stationary phase but can be eluted with reversed-phase eluents [63].

Steevensz et al. [63] analyzed phlorotannins’ extracts from five plants by using ultra-high-pressure liquid chromatography (UHPLC) which makes use of hydrophilic interaction liquid chromatography (HILIC) mode combined with high-resolution mass spectrometry (HRMS) operation. Among them, the sample was taken from the sac of *Fucus vesiculosus*, and the separation was achieved in less than 15 min. The alkaline mobile phase enhances negative ion electron-spray ionization (ESI), and produces multiply charged ions that allowed the detection of high molecular weight phlorotannins. Agregán et al. [64] analyzed phenolic compounds from *Ascophyllum nodosum* (ANE), *Bifurcaria bifurcata* (BBE) and *Fucus vesiculosus* (FVE) by using HPLC-diode array detection combined with negative electron spray ionization tandem mass spectrometry (LC-DAD-ESI-MS/MS) and their potential applications as functional constituents were evaluated. Phlorotannins were preliminarily identified as the main phenolic compounds in the three extracts, followed by phenolic acids and flavonoids. HRMS enabled the identification of phlorotannins with masses up to 6000 Da using a combination of accurate mass and ^13^C isotopic patterns.

### 4.4. Nuclear Magnetic Resonance

Due to the instability of the phlorotannin, and to make NMR analysis easier, these compounds are often acetylated by using acetic anhydride and pyridine [65]. Acetylation of the phenolic hydroxyl group prevent the ketone–enol tautomerism, thereby inhibiting oxidation. In these types of phenolic compounds, the hydrogen between two phenolic groups on the aromatic ring is susceptible to the exchange of hydrogen deuterium with solvents such as deuterium oxide and deuterated methanol, which can lead to a reduced or complete exchange of peaks in the spectrum. However, acetylation of phlorotannins prevents this exchange from occurring. Before identifying compounds by NMR spectroscopy, they must first be purified by chromatography. Acetylation of phlorotannins alters the polarity of the compound, so once acetylated, phlorotannins can be separated by normal phase silica chromatography using solvent systems of n-hexane, chloroform, methanol or ethanol [66]. ^1^H NMR has been used to confirm the presence of phenolic components as well as to detect [67] and quantify the phlorotannins [68,69]. Protons from phenolic units have characteristic chemical shifts that can be used to detect the presence of phenolic moieties.

Anaëlle et al. [38] tracked phenolic compounds in the range of 5.5 to 6.5 ppm by high-resolution magic-angle rotational nuclear magnetic resonance (HR-MAS) and mannitol (common co-extracts in phenolic extracts) at 3.6–3.9 ppm (polyol region), and analyzed the relative ratios of the two. Other studies on brown algae have also used ^1^H NMR, tracking is performed in the range of 5.5–6.5 ppm to confirm the presence and relative quantity of phenolic compounds [70]. Moreover, the methods that may be used from pretreatment to chemical characterization of brown seaweeds’ polyphenols are summarized below (Figure 3).

## 5. Applications of Phlorotannins

Despite the diversification of pathways discovered by pharmaceutical companies in recent years, natural products, especially medicinal plants, remain an important source of drugs for new chemical entities. Natural products contribute significantly to the treatment of many diseases: cancer, acquired immune deficiency syndrome (AIDS), Alzheimer’s, malaria and pain. The diverse structure of natural products and the characteristics of easy binding with biological macromolecules determine their incomparable advantages in participating in the physiological process of life, which give natural products an irreplaceable and important position in the development of new drugs. They are also important sources for discovering candidate drugs and drug pilot structures. There is a type of very unique natural active substance in brown seaweeds: the phlorotannins, which are found in neither green nor red seaweeds. As a medicine of natural origin derived from green plants, it has no drug residue and causes no pollution in animals. Phlorotannins derived from brown seaweeds are natural substances that are easily absorbed and utilized by the human body, and the parts that are not absorbed can be smoothly excreted with the metabolic process of the substance. At the same time, its toxic side effects are small or even non-existent, and the animal body does not develop drug resistance. In recent years, research on the biological activity and application of phlorotannins has been one of the hot topics in marine drug development (Table 7).

The ecological environment of marine organisms is significantly different from that of terrestrial organisms. Marine organisms contain a large number of substances with novel structures and extremely strong biological activities, so marine natural products are one of the most active fields in natural product chemistry research. The characteristics of marine compounds are summarized as follows:

First, there is a large amount of halogen-containing organic matter. The most important characteristic of marine natural products is the composition of the elements. Because seawater is rich in halogens, marine organisms contain many covalently bound halogen-containing organic compounds, sometimes the amount of halogen atoms can reach more than 70% of the total number of elements in the molecule, the most common is bromine, followed by chlorine and iodine. The presence of polyhalogeno is specific to marine natural products, while organo-bromides and organo-iodides have not been found in the metabolites of land-based organisms to date;

Second, the structure is unique, complex and changeable and biologically active. Some marine toxins are the defense substances of marine organisms, because they are quickly diluted by seawater after release, in order to achieve the role of defense, their activity is often super, such as palytoxin (PTX, from *palythoa toxica*) and tetrodotoxin (TTX, from *tetraodontidae*);

Third, the structure contains rare or special groups, marine biological secondary metabolites not only contain a unique structural skeleton, but also contain some special substituents. Such as isocyanato, guanidine, cyclic polysulfide, peroxyl group and rare diallyl or ethynyl in nature. The discovery of these rare groups in marine natural products has opened scientists’ eyes and expanded the range of diverse groups present in natural ingredients.

We also summarized the advantages and disadvantages of some of the substances [95,96] based on marine compounds in Table 8. 

### 5.1. Applications in the Field of Medicine

Phlorotannins, as a role of plant polyphenols, based on its hydrophilic and hydrophobic moieties, can interact with plasma proteins in the blood by non-covalent interactions (e.g., multiple hydrogen bonding, electrostatic, and cation-π interactions) [97,98]. Based on phenol units, polyphenols have the innate ability to reduce and scavenge free radicals and have a high affinity for proteins through specific or non-specific interactions [99]. They can interact with several receptors, regulate signal transduction, regulate enzyme activity, inactivate microorganisms and cross-link biological macromolecules [100], which indicates broad application prospects in tissue engineering. In the same way as the tannins of terrestrial plants, these phenolic compounds are highly soluble in water, strongly bind to proteins, polysaccharides, and other biopolymers, chelate divalent metals and have a polymer structure. Polyphenolic compounds extracted from seaweeds are bioavailable. Polyphenolic compounds can be absorbed either directly in the upper digestive tract unchanged or in the lower intestine after being modified by the bacteria present there [101].

Related studies have shown that phlorotannins from brown seaweeds showed low toxicity in cell lines, microalgae, seaweed spores, plants, invertebrates, animals (fish, mice, rats, and dogs) and humans at a moderate dosage [102]. Mild side-effects were recorded in humans, fish and dogs. However, the safety and toxicity of phlorotannins in aquaculture fish, livestock and companion animals are limited. These findings can serve as a basis for developing these compounds into novel functional foods, animal feeds and drugs. Some clinical studies on the use of brown seaweeds’ polyphenols in humans are shown in Table 9.

#### 5.1.1. Cancer Prevention and Treatment

Excessive oxidative stress can cause oxidative damage to intracellular biological macromolecules, promote abnormal gene expression and DNA structural changes and block vital signaling pathways between cells [107]. Brown algae polyphenols have good anticancer activity, and their mechanism of action is mainly manifested either directly as a pro-apoptotic, antiproliferative, anti-metastatic or anti-angiogenic agent, or indirectly by inhibiting the oxidative-stress inflammatory network closely related to tumorigenesis [108]. The anticancer activity of brown algae polyphenol crude extracts differs greatly from purified isolated components. In mouse models containing sarcoma xenografts, the in vivo antitumor effect of eckol was shown not only due to its ability to interfere with the expression of caspase-3, caspase-9, Bcl-2 and Bax genes, but also due to its ability to inhibit epidermal growth factor receptor (EGFR) expression. Thus, in addition to its antiproliferative properties, eckol exerts antitumor activity by stimulating the host immune response. Imbs et al. [109] found that among three different seaweeds, *Fucus evanescens*, *Laminaria cichorioides* and *Costaria costata*, the former with 60% ethanol extracts have highest content of polyphenols (10.1% dry matter), showed the strongest inhibitory effect on DLD-1 and HT-29 colorectal adenocarcinoma cell growth (67% and 63%).

Some studies have shown that, in the same way as plant phenolics, phlorotannins tend to degrade during their passage throughout the gastrointestinal tract, and only a small portion will reach the intestinal lumen intact and become bio-accessible for further absorption [89]. According to the work of Ahn et al. [110], carried out in xenograft mice models implanted with SKOV3 ovarian cancer cells, the oral administration of dieckol (300 mg/kg/week) was even more effective than cisplatin (9 mg/kg/week) at suppressing the tumor growth without showing any liver or kidney toxicity, while the cisplatin-treated mice revealed increased blood urea nitrogen and serum creatinine which are indicative of kidney dysfunction. *Dieckol*, derived from edible brown alga *Ecklonia cava*, can significantly enhance the inhibition of tumor growth by cisplatin with lower weight loss and kidney damage in a mouse model [111]. It also indicated that brown algae phlorotannins may improve the efficacy of platinum drugs for ovarian cancer by enhancing cancer cell apoptosis via the ROS/Akt/NFκB pathway and reduce nephrotoxicity by protecting against normal kidney cell damage.

#### 5.1.2. Treatment of Neurodegenerative Diseases

The phenomenon of excitatory toxicity caused by glutamate is associated with the pathophysiology of a variety of central nervous system diseases, which can lead to neuronal dysfunction, degeneration and apoptosis [112,113,114]. Brown algae polyphenols have been found to protect brain cells from glutamate excitatory toxicity through multimodal action. The mechanism of action is as follows: the ability to inhibit central nervous system (CNS)-related enzymes (acetylcholinesterase and butyrylcholinesterase, monoaminoxidase, β-secretase, tyrosinase); Regulation of neuronal receptors; Regulation of the signaling pathways and neuroinflammation associated with oxidative stress-mediated neuronal cell death [2].

Yang et al. [115] demonstrated that phloroglucinol, the basic unit of brown algae polyphenols, regulates synaptic plasticity. With a decrease in dendritic spine density and synaptic protein levels, cognitive dysfunction in 5XFAD mouse models was alleviated. After oral administration of it, a significant decrease in the number of Aβ plaques and the level of the BACE-1 protein was also observed. In addition, resorcinol prevents lipid peroxidation, slows the reactivation of glial cells, and reduces the release of pro-inflammatory cytokines in 5XFAD mice.

Polyphenols such as phlorotannins must cross the physical barrier of the CNS to exert a neuroprotective effect: the blood–brain barrier (BBB) that separates the circulating blood from the brain extracellular fluid. Phlorotannin’s action through the BBB on gamma aminobutyric acid type A (GABAA)-benzodiazepine receptors has been demonstrated. *Dieckol* [116] and *eckol* [117] have been effectively shown to successfully penetrate the brain by the BBB via still unknown transportation mechanisms.

#### 5.1.3. Development of Novel Antifungal Drugs

Polyphenolic compounds have a universal inhibitory effect on microorganisms [102]. On the one hand, phlorotannins block the synthesis of dimorphic complexes in fungal cells, resulting in an altered appearance and surface adhesion properties of *Pseudomonas*, thereby reducing the toxicity and ability of pathogenic fungi to invade host cells. On the other hand, phlorotannins induce reactive oxygen species (ROS) production and trigger early apoptosis, leading to activation of the CaMCA1 gene and destruction of cell membranes in Candida albicans. These inhibitory effects reflect the potential of phlorotannins as a novel antifungal drug [44,118,119].

Lopes et al. [120] crushed the dried *Fucus spiralis Linnaeus*, degreased with n-hexane and extracted with acetone: water (7:3). The extract is purified with cellulose and then washed with toluene. Then, the cellulose is washed with acetone: water (7:3) to obtain phlorotannins. These phlorotannins exhibit antifungal activity against *Candida albicans*, *Aspergillus* and *Dermatophytes*. The MIC values of phlorotannins for these fungi range from 3.9 to 31.3 mg/mL.

At present, most of the antibacterial activity studies on brown seaweeds’ polyphenols are in vitro activity studies, and no pharmacokinetic analysis of polyphenols exerting antibacterial effects has been found.

#### 5.1.4. Development of New Blood Pressure Lowering Drugs

Angiotensin-converting enzyme (ACE) is an ideal target for the treatment of diseases such as hypertension, heart failure, diabetes mellitus and hypertension. In the study of blood pressure-lowering activity of brown algae, it has been proved that brown algae such as *Undaria pinnatifida*, *Scagassum*, *Laminaria japonica*, *Fucus vesiculosus* and *Hizikia fusifarme* contain blood pressure-lowering active substances [121].

Dieckol isolated from the ethanol extract of *Kombu* is a non-competitive inhibitor against ACE I that induces the production of nitric oxide (NO) in EA.hy926 cells and has no cytotoxic effect [85]. Ko et al. [122] extracted 6,6’-bisphenols from the *Ecklonia cava* and studied its biological activity, finding that ACE inhibitory activity was formed by the interaction of hydrogen bonds and Pi bonds between ACE and 6,6’-biphenol. The 6,6’-bisphenol significantly increased the content of NO by phosphorylation of nitric oxide synthase in human endothelial cells.

Until now, most of these ACE inhibitory activities have been observed in vitro or in mouse model systems. Therefore, further research studies are needed in order to investigate their activity in human subjects. So far, there were no clinical studies of phlorotannins in patients with hypertension.

#### 5.1.5. Anti-Diabetic and Anti-Obesity Potential

In recent years, with the separation and identification of active ingredients in *Laminaria japonica*, *Undaria pinnatifida*, *Scagassum* and other brown algae, it has been proved that most brown algae contain hypoglycemic active substances [123]. Phlorotannins from the genus kombu exhibit anti-diabetic activity mainly through the inhibition of α-amylase, α-glucosidase, pancreatic lipase and aldose reductase (AR), which can promote the delay of carbohydrate and lipid digestion, followed by reducing postprandial plasma glucose levels and overall body weight, thereby helping to prevent and improve metabolic disorders such as diabetes mellitus type 2 (T2DM) and obesity [124].

Phlorotannins isolated from methanol extracts of *Ecklonia stolonifera* have inhibitory activity against α-glucosidase, which is attributed to the presence of *rhizolutin A*, dieckol and *7-rhizophericol* [125]. Lee et al. [126] extracted a phlorotannin from the *Ecklonia cava*, modeled on streptozotocin-induced diabetic mice, and after feeding, blood samples were taken from the tail vein at 0, 30, 60 and 120 min and the blood glucose amount was measured. It was found that it had strong inhibitory activity against α-glucosidase and α-amylase.

Further clinical studies have shown that the physiological effects of seaweed supplementation can reduce fasting blood glucose and two hours postprandial blood glucose levels in patients with type 2 diabetes (control: 254.4 ± 22.8 mg/dL, seaweed supplementation: 203.1 ± 12.3 mg/dL) without affecting glycated hemoglobin levels [127]. Shannon and Abu-ghannam [128] found that the daily supplementation of *Undaria pinnatifida* balanced the blood glucose levels in patients with type 2 diabetes mellitus by clinical trial.

A 24-week randomized, double-blind, placebo-controlled crossover trial was conducted to investigate the effects of brown seaweeds’ polyphenols on DNA damage and antioxidant activity in overweight or obese people [105]. A total of 80 participants (BMI units: kg/m^2^ ≥ 25) aged 30–65 years consumed either a 400-mg capsule containing 100 mg seaweed (poly)phenol and 300 mg maltodextrin or a 400-mg maltodextrin placebo control capsule daily for an 8-wk period. Consumption of the seaweed (poly)phenols resulted in a modest decrease in DNA damage but only in a subset of the total population who were obese. There were no significant changes in C-reactive protein (CRP), antioxidant status or inflammatory cytokines.

#### 5.1.6. SARS Virus Inhibitors

Due to its multiple functions and necessary roles in viral replication and pathogenesis, cysteine proteases can be used as an ideal target for antiviral drugs. Among them, the master protease (3CL) is a structural protein that affects the survival and reproduction of the virus by cutting the site of action of the polymer protein. The 3CL protease is also an enzyme necessary for coronavirus replication, which is a target for potential anti-SARS drugs. Of the nine phlorotannins isolated from the ethanol extract of the *Ecklonia cava*, dieckol, which is connected by diphenyl ether, exhibits the most potent SARS-CoV inhibitory activity. In addition, dieckol has the most potent inhibitory activity against cell-based 3CL, which is more potent than other phlorotannins’ derivatives and natural reference inhibitors. Dieckol has a high association rate with proteins and forms strong hydrogen bonds with the catalytic binary of SARS-CoV 3CL (Cys145 and His41) and consumes the lowest binding energy [129].

To explore the pharmacokinetics of brown macroalgal polyphenols during their antiviral activity, Gunaseelan et al. [78] evaluated the potential of root tannins as multiple-target protein antagonists necessary for SARS-CoV-2 replication. Top-ranked marine brown algal phlorotannins are difucol hexaacetate, diphlorethol pentaacetate, eckol hexaacetate and fucofuroeckol.

Based on the existing studies of ADMET (Absorption, Distribution, Metabolism, Excretion and Toxicity) characteristics of phlorotannins’ compounds, have clearly comprehended the bioavailability of the phlorotannin ligand molecules as an oral drug candidate [78]. Followed by the first-pass metabolism, polarity, aqueous solubility and permeability of the gut–blood barrier appeared at an adequate level as compared with standard drugs. Furthermore, the metabolic reaction and blood–brain barrier are also observed under limitation as per the standard volume, which can be seen in moderate levels in all of the ligands except phlorofucofuroeckol A and B, fucodiphloroethol G, tetrafucol A, tetraphlorethol, fucotriphlorethol and diphlorethohydroxycarmalol.

Comparative studies of the route of administration were conducted by analyzing the results obtained from standard drugs used to combat COVID-19, such as remdesivir, ritonavir, favipiravir, lopinavir and hydroxychloroquine. Among them, hydroxychloroquine showed the highest human oral adsorption rate (about 91%), while the most commonly used remdesivir showed only 36% oral adsorption against COVID-19. The dioxinodehydroeckol, diphlorethol, diphlorethol pentaacetate and difucol hexacetate manifested a higher degree of oral absorption (58.0, 58.05, 58.84, 47.30, respectively) as compared with standard remdesivir [130]. At the same time, diphlorethol pentaacetate (70.56) had a higher Madin-Darby canine kidney (MDCK) permeability within the given range (25 to 500) than standard remdesivir (just 6).

#### 5.1.7. Metabolomic Profiling Analysis of the Extract of Brown Seaweeds

The majority of phlorotannin metabolites were found in samples collected at late time points (6–24 h), indicating limited small intestinal absorption. Seaweed phlorotannins are metabolized and absorbed, predominantly in the large intestine, and there is a large inter-individual variation in their metabolic profile [131]. In the upper GI tract, dietary polyphenols act as substrates for a number of enzymes, and they are subjected to extensive metabolism by glucosidase enzymes, phase I enzymes (hydrolyzing and oxidizing) [132]. Thus, phlorotannin intake results in the formation of phase II conjugate metabolites (glucuronides, sulphates). Further transformations can occur in the colon, in which the enzymes of the gut microbiota act to breakdown complex polyphenolic structures to smaller units, which may also be absorbed and further metabolized in the liver [133].

In recent years, brown algae polyphenols have been a research hotspot in the field of marine compounds. However, there have been few studies into the metabolic changes following ingestion of brown algae extract. Kim et al. [134] investigated the metabolic effects of a 12-week consumption of *E. cava* polyphenol extracts (seapolynol) in subjects exposed to moderate caloric intake and physical activity and with a body mass index (BMI) higher than 25 kg/m^2^ and lower than 30 kg/m^2^. Urinary metabolomic profiling analysis showed that the levels of riboflavin, urocanic acid, 5-hydroxy-6-methoxyindole glucuronide and guanidino valeric acid were significantly increased in the seapolynol intake group compared with the placebo group. These findings suggest that the decreased body fat induced by the intake of seapolynol is related to an increase in the antioxidant effect of riboflavin.

### 5.2. Applications in the Food Sector

#### 5.2.1. As a Functional Food against Hypercholesterolemia

The main risk factor for coronary heart disease is excessive dietary cholesterol depletion, which, once arterial plaques form, can lead to fatal myocardial infarction, heart attack and cerebrovascular disease [135]. Currently, the intake of naturally active products can be used as a strategic therapy for the combined use of anti-hyperlipidemia drugs [136]. Studies have shown that algae compounds have the ability to bind dietary cholesterol, and brown algae have the ability to accelerate the excretion of cholesterol by the human body. Therefore, brown algae belong to a valuable food source of natural compounds that can be used to treat high cholesterol. A variety of brown algae species have shown the ability to lower blood lipid levels and total cholesterol [137].

Phlorotannins regulate serum lipid levels in most cases, reducing the risk of cardiovascular and cerebrovascular disease. The hypolipidemic effects of phlorotannins may also be associated with the antioxidant, anti-inflammatory and hepatoprotective activity of polyphenol compounds, thereby inhibiting fat-induced liver damage and ensuring lipid normalization [138,139]. Dieckol isolated from *Fucus vesiculosus* and *Ecklonia cava* has been reported to have the ability to inhibit HMG-CoA reductase, affecting cholesterol synthesis [140,141], thus demonstrating that phlorotannins have the same mode of action as statins.

#### 5.2.2. As Food Antioxidants

The special phenolic hydroxyl structure endows brown algae polyphenols with antioxidant activity [142]. The phenolic hydroxyl structure can provide hydrogen donors and scavenge a variety of reactive oxygen species (ROS). On the one hand, the excited state oxygen molecules are reduced to the ground-state oxygen with lower activity, and the generation of oxygen free radicals is inhibited. On the other hand, as a free radical scavenger, brown algae polyphenol can combine with oxygen free radicals, reduce the activity of free radicals and hinder the chain-reaction of free radical scavenger [143]. The strength of the antioxidant effect of polyphenols is affected by the molecular structure and the number of hydroxyl groups, chain length and number of intramolecular hydrogen bonds [144,145]. Taking advantage of the antioxidant effects of brown algae polyphenols, it can also be used to treat degenerative diseases such as cardiovascular disease, diabetes, cancer, atherosclerosis and Alzheimer’s disease [146].

Meng Tong [147] studied the effect of polyphenols from *Laminaria japonica* on the quality of emulsified intestines, and found that the brown algae polyphenols can reduce the content of carbonyl groups and the loss of sulfhydryl groups, and inhibit the oxidation of proteins [148]. In addition, when brown algae polyphenols are used to preserve oils and fats, they can act as antioxidants and color protection at the same time. Brown algae polyphenols can also undergo complex reactions with metal ions, thereby inhibiting the oxidation of food by a small amount of metal ions.

#### 5.2.3. As Food Preservatives

Brown algae polyphenols are a kind of extract with both antibacterial and antioxidant activities, which can be directly used in the preparation of new edible packaging films. Studies have shown that this packaging film has the functions of antibacterial preservation and prevention of physical damage, and has played a great role in the field of food storage and preservation [149]. Low molecular weight phlorotannins extracted from *Sargassum thunbergia* damaged the cell membrane and cell wall of *Vibrio parahaemolyticus*, causing cytoplasm leakage and deconstruction of membrane permeability.

Shi [150] prepared nanoparticles embedded with algal polyphenols by the ion gel method, and found that the preservative obtained by this method had a better preservation performance and longer preservation time. Some phlorotannins derived from brown algae species have been studied in the application of antibacterial agents (Table 10).

#### 5.2.4. As Pesticides

The larvicidal effects of phlorotannins are mediated by multiple mechanisms, including direct inhibition of the settlement and/or survival of larvae and regulation of the growth of bacterial micropollutants, thereby affecting larval settling. Phlorotannins isolated from *Sargassum* inhibited the metamorphism of 33% of *Ciona Savignyi* and 27% of *Halocynthia roretzi* at low concentrations (25 μg/mL). These findings suggest that phlorotannins can act as antifouling agents without causing damage to other organisms [75].

The larvicidal activities of phlorotannins in mosquitoes reported by Ravikumar et al. [154] and Manilal et al. [155] suggest that they may be effective repellents. Phlorotannins have an effect on the larvae of marine invertebrates [156], suggesting that they are natural antifouling agents. Unlike heavy metals, which act as broad-spectrum toxins against both target and non-target marine organisms, the natural antifouling effect of phlorotannins is specific to target organisms.

### 5.3. Application in the Field of Cosmetics

#### 5.3.1. Whitening and Beauty Effects

Tyrosinase is a key enzyme in the process of melanin formation. By adding tyrosine inhibitors to cosmetics to inhibit the formation of melanin, the whitening effect of cosmetics can be achieved. Melanin production begins by oxidation of tyrosine to dopaquinone, which is catalyzed by tyrosinase (TYR). The tyrosinase-associated protein-1 (TRP-1) and TRP-2 or dopachrome tautomerase also play an important role in all eumelanin-producing reactions [157].

Kang et al. [158] extracted and isolated five kinds of polyphenols from *Ecklonia stolonifera*. Among them, phlorofucofuroeckol A could significantly inhibit the activity of tyrosinase. *Eisenia arborea* phenols isolated from the polyphenolic compounds of the brown alga *Ecklonia cava* can significantly inhibit tyrosinase activity and prevent the formation and accumulation of melanin. Further research found that 974-A, phlorofucofuroeckol-A and eckol, isolated from *Ecklonia stolonifera Okamura*, reduced the cellular melanin content and tyrosinase activity, and downregulated the expression of melanogenesis enzymes including tyrosinase, tyrosinase-related protein (TRP)-1 and TRP-2 in B16F10 melanoma cells [81]. These compounds also effectively scavenged radicals at the cellular level.

#### 5.3.2. Treatment of Atopic Dermatitis

When the body is attacked by harmful stimuli or pathogens, the body’s immune system initiates an inflammatory response to remove the harmful substances and protect itself from harm. Atopic dermatitis (AD) is a cutaneous manifestation of a systemic disorder that can lead to asthma, food allergies and allergic rhinitis [159].

Kim et al. [160] studied the activity of kombu ethanolic crude extract and found that the purified eckol can downregulate the expression of pro-inflammatory factors, tumor necrosis factor and two interleukins in mouse macrophage leukemia cells, confirming that brown algae polyphenol compounds have good anti-inflammatory activity and can be used to treat inflammation. A brand-new phlorotannin was isolated from *Ecklonia kurome Okam*, and the in vitro study of phlorofucofuroeckol-B on rat basophilic leukemia (RBL)-2H3 cells confirmed that this tannin is able to inhibit histamine release, thus guaranteeing anti-allergic properties [161].

#### 5.3.3. Matrix Metalloproteinase Inhibitors

Matrix metalloproteinases (MMPs) are able to digest extracellular matrix components such as collagen, proteoglycans, fibronectin and laminin in vitro and in vivo. In particular, gelatinase, which effectively cuts collagen types IV and V, can degrade collagen and elastic fibers, resulting in loss of skin elasticity, promoting wrinkle formation and accelerating aging. Some studies have suggested that the effectiveness of the dieckol in downregulating the expression of MMPs preventing the cellular invasion. For example, dieckol treatment in HT-1080 cells has reduced the intracellular ROS levels, inhibited the activation of Rac1 along with expression of focal adhesion kinase (FAK) and prevented the expressions of MMP-2, MMP-9 and MMP-13 [162].

Two phlorotannins, isolated from methanolic extracts of the marine brown alga, have been reported to inhibit the protein and gene expression levels of MMP-1, MMP-3 and MMP-13 in human osteosarcoma cells (MG-63) [163]. Dieckol and Eckol, isolated from *Laminaria spp*, inhibited the expression of MMP-1 in human dermal fibroblasts [164]. These results suggested the phlorotannin could promote cell differentiation, attenuate MMP-1, MMP-3 and MMP-13 expressions, and inflammatory responses via the MAPK pathway in chronic articular diseases.

#### 5.3.4. As UV Sunscreens

Under ultraviolet radiation, the human body can produce a variety of effects, such as DNA damage, inhibition of DNA replication or mutations, photosynthetic apparatus impairment, decrease in CO_2_-fixation, production of (ROS) [165]. Meanwhile, a decrease in the degree of photo-inhibition is commonly observed in brown algae. Such an effect may be explained either by activation of the antioxidative response, or by the formation of UV-screening compounds [166] and an increase in the activity of repairing enzymes. Phlorotannins are able to absorb UV radiation, mainly UV-C and partly UV-B, with maxima at 195 nm and 265 nm making them good candidates for UV protection [167,168].

Gómez et al. [169] found that the induction of phlorotannins during UV exposure can alleviate the inhibition of photosynthesis and DNA damage in the kelp *Lessonia nigrescens*, two major detrimental effects of UV. Of course, the use of phlorotannins as a sunscreen is not only due to the anti-ultraviolet radiation effect, which requires antioxidant, anti-inflammatory and antibacterial effects to work synergistically. The ethyl acetate fractions that were isolated from the brown alga *Polycladia myrica* have a greater ability to inhibit free radicals as well as inhibit the growth of Gram-positive bacteria. Soleimani et al. [170] used the ethyl acetate fraction (as a biofilter) at a concentration of 5% as F3 and studied its sun protection efficacy, physical properties and stability. It turned out that the formulation F3 with SPF = 31.8 ± 4.7, UVA/UVB ratio = 0.98, showed excellent UVR protection, compared with commercial sunscreen (SPF = 29.76 ± 5.5, UVA/UVB ratio = 0.95). During stability studies, cream was formulated with a 5% fraction of brown algae without changes in appearance and pH.

## 6. Hindrance of Phlorotannins in Application

Phlorotannins isolated from seaweed are compounds with high biological activity, and have broad development potential in the fields of medicine, food and health products. However, polyphenols are compounds with complex structures and less content in brown algae, and how to obtain extremely pure raw materials and maximize extraction of active ingredients is the first hurdle to overcome. The polyphenolic compounds in brown algae are complex and variable, and no library or standards are available for the structural characterization of root-bark tannins. Second, the impact of heavy metal residues and bio-enrichment of toxic substances on brown algae raw materials should be considered due to marine pollution (marine debris, oil spills, nuclear leaks, et al.) in recent years. Third, how to control the solvent residue in the application process is also a focus that needs attention. Currently, the extraction methods for phlorotannins are mainly the organic solvent extraction method, such as ethanol, methanol, acetone and ethyl acetate which are commonly used extraction solvents, which exceeds a certain limit may cause damage to animals and humans. In addition, the bioavailability and metabolomic analysis of phlorotannins are very limited, which poses a significant obstacle to clinical studies. There is still a very long and difficult way to go from chemically active substances to pharmaceutical preparations, which makes it difficult to launch new drugs based on phlorotannins as the main raw material.

## 7. Conclusions

Brown algae polyphenols have a wide range of biological activities and have broad research prospects. However, due to the complex structure of brown algae polyphenols, it is difficult to obtain completely purified polyphenolic substances. Researchers have also encountered many obstacles in their research on brown algae polyphenols, most of which are only based on the extraction process and biological activity of brown algae polyphenols, and lack of research on the molecular structure, mechanism of action and structure–activity relationship. For example, the structural identification, structure–activity relationship and mechanism of action of brown algae polyphenols with hypoglycemic activity are not clear for the time being. With the continuous development and progress of omics, it is necessary to continue to deepen the study of brown algae polyphenols, a natural marine product, and make full use of the rich brown algae resources and a variety of biological activities to provide reference for the vigorous development of marine resources.

## 8. Materials and Methods

In this review, we conducted a PubMed search to cover all the available studies on the experimental extraction, purification, identification, classification, in vitro and in vivo active effects of isolated phlorotannins and phlorotannin-rich extracts/fractions. The query terms used for the PubMed database search included the terms “phlorotannins”, “brown seaweed”, “polyphenols”, “structure”, “extraction”, “characterization”, “purification”, “bioavailability”, “cancer”, “tumor”, “antioxidant”, “neurodegenerative”, “antifungal”, “anti-bacterial”, “ACE”, “hypoglycemic ”, “virus”, “tyrosinase”, “ inflammatory”, “MMPs”, etc. Additionally, this search was complemented by further exploring the references of the articles retrieved from the PubMed search. All references are studies on marine polyphenols published in the last two decades (from 1996 to 2022).

## Figures and Tables

**Figure 1 marinedrugs-20-00742-f001:**
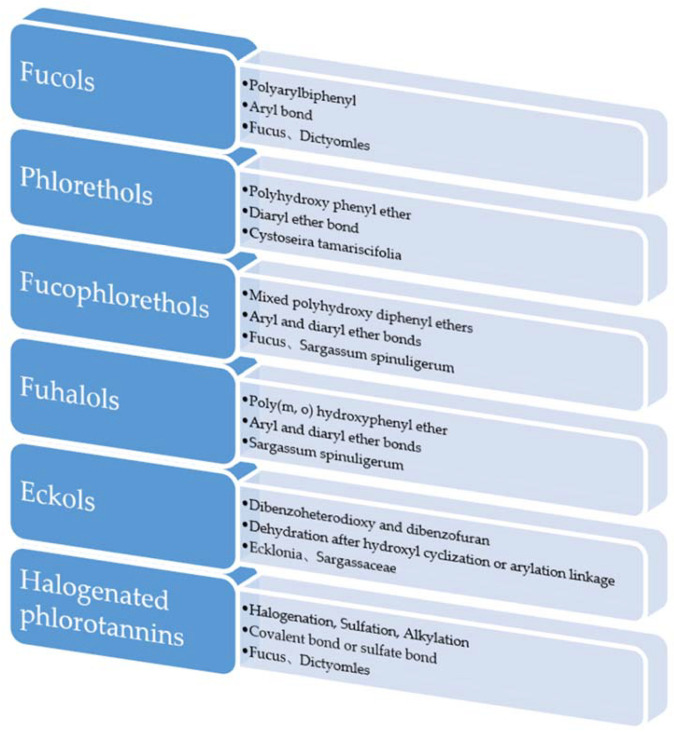
Classification of small molecular-weight brown algae polyphenols.

**Figure 2 marinedrugs-20-00742-f002:**
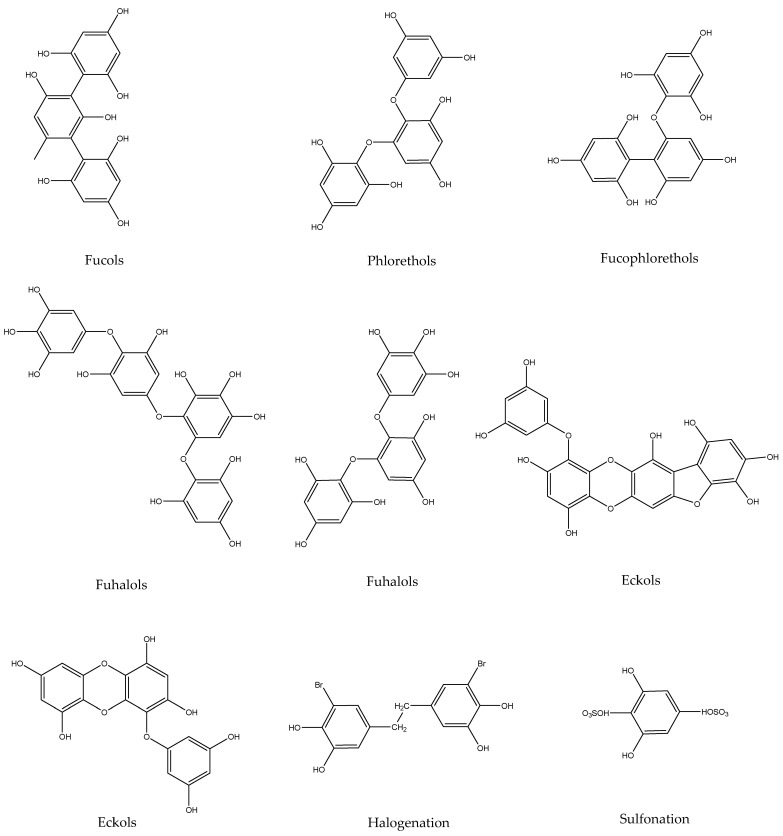
Chemical structure types of phlorotannins in brown algae.

**Figure 3 marinedrugs-20-00742-f003:**
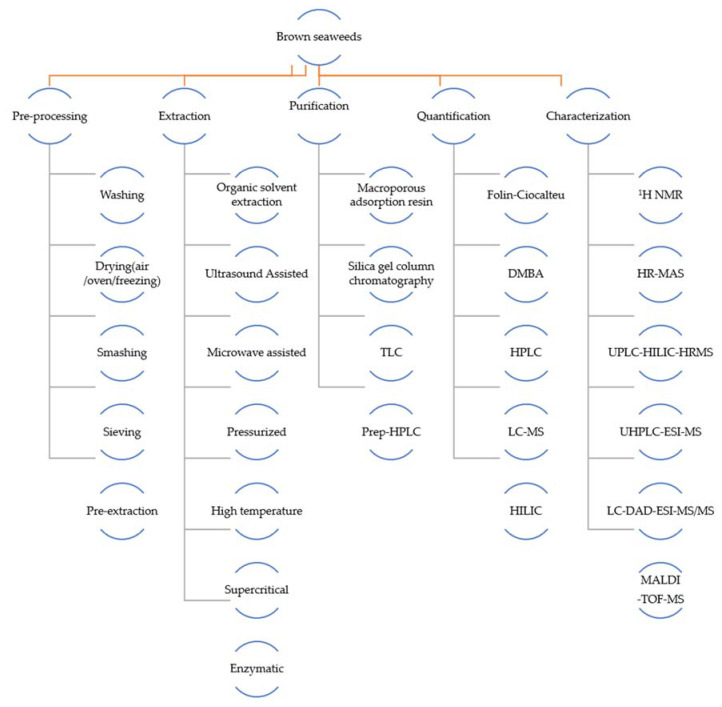
Possible methods for obtaining polyphenols from brown seaweeds.

**Table 3 marinedrugs-20-00742-t003:** Microwave-assisted extraction (MAE) of marine phenolics.

Seaweed Type	State of the Seaweed (Wet/Dry/Particle Size)	Organic Solvents Used	Solvent–Seaweed Ratio/Temperature(°C)/Time (min)/Microwave Output Power(w)	Yield	Application of the Extract	Reference
*Sargassum swartzii*	Moisture content was less than 10%; grinding and cryopreservation at 4 °C	Ethanol concentration of 52%	33:1/-/65 min/613 w	27.88 ± 0.13 mg/g	Potential antioxidant	[30]
*Carpophyllum flexuosum*	Dried (oven, 60 °C, 24 h) and milled to 1 mm and stored at −20 °C in vacuum-sealed bags	Water	30:1/160 °C/3 min/-	11.4% (PGE, % of DW)	-	[16]
*Carpophyllum* *flexuosum*	Powder through a 1.0 mm sieve stored at −20 °C in sealed bags	Milli-Q water	30:1/160 °C/3 min/-	15.8 ± 0.3 (PGE % of DW)	Antioxidant activities	[18]
*Carpophyllum plumosum*				9.2 ± 0.6 (PGE % of DW)		
*Ecklonia radiata*				2.0 ± 0.1 (PGE % of DW)		
*Fucus vesiculosus*	Particles lower than 0.8 mm; stored under vacuum (75%) at −20 °C	57% (*v*/*v*) ethanol	10:1/75 °C/5 min/-	9.8 ± 1.8 mg/g	Potential antioxidant	[31]
*Saccharina japonica*	Dried powder through a 40-mesh sieve	55% ethanol	8:1/60 °C/25 min/400 W	0.585 mg PGE/g DW	Inhibition of HepG2 cancer cells	[32]
*Polysiphonia, Ulva* and *Cladophora (mixed)*	Through a 0.25-mm sieve, and dried in an oven at 45 °C	Milli-Q water	3:1/25 °C/30 min/1000 w	544 mg/L	Antibacterial properties	[33]
*Ascophyllum nodosum*	/	Methanol: water (70:30)	10:1/110 °C/15 min/-	/	α-glucosidase inhibitory activity, DPPH, ABTS scavenging ability	[34]

**Table 5 marinedrugs-20-00742-t005:** Supercritical fluid extraction (SFE) of marine phenolics.

Seaweed Type	State of the Seaweed (Wet/Dry/Particle Size)	Organic Solvents Used	Solvent–Seaweed Ratio/Flow Rate/Co-solvent/Pressure/Temperature	Yield	Application of the Extract	Reference
*Undaria pinnatifida*	Powder	Ethanol modified CO_2_	2 mL·min^−1^/ethanol/250 bar/333 k	800 μg/g	-	[41]
*Sargassum muticum*	comminuted (<0.5 mm)	Pure scCO_2_	25 g CO_2_ ·min^−1^/10% methanol/20 MPa/50°C/1 h	4% of the dry weight	Anti-browning activity on B16F10 murine cells and inhibition of lipogenesis in SW872 liposarcoma cells	[42]
*Saccharina japonica*	Dried at 80 °C for 72 h and sifted through a 710-mesh sieve	Supercritical carbon dioxide (SC-CO_2_)	1:1/27 g·min^−1^/2.00% Sunflower Oil (SFO) and water/300 bar/48.98 °C/2 h	0.927 ± 0.026 mg/g	Antioxidant activities	[43]
*Undaria pinnatifida* *Laminaria digitata*	Powder	Supercritical carbon dioxide	34 kg CO_2_·h^−1^/-/37.9 MPa/50°C/-	/	Against fruit post-harvest fungal diseases	[44]

**Table 7 marinedrugs-20-00742-t007:** Bioactivities of Phlorotannins.

NO.	Bioactivity	Reference	Mechanism of Action	Significant Findings
1	Antioxidant activity	[71]	Radical scavenging activity	Phlorotannins from marine algae, including *Ecklonia cava*, have been shown to protect cells from radiation-induced injury as well as oxidative stress.
2	Anti- inflammatory activity	[72,73]	Direct scavenging nitric oxide (NO); Decreasing NO production through the inflammatory signaling cascade and inhibiting the enzymes involved in NO production.	Phlorotannins were found to inhibit TNF-α, IL-1β and PGE2 production at the protein levels. The anti-inflammatory properties of our compounds are related to the downregulation of proinflammatory enzymes, NOS and COX-2, through the negative regulation of the NF-κB pathway in Aβ25–35-stimulated PC12 cells.
3	Antibacterial activity	[74,75]	Block dimorphic complexes, resulting in the appearance of pseudo hyphae with decreasing surface adhesive properties. Decrease the virulence and capacity to invade fungal host cells; Induced reactive oxygen species (ROS) production and triggered early apoptosis, resulting in the activation of the meta-caspase CaMca1 and membrane disruption.	Phlorotannins from brown seaweeds show antifungal activity against dermal and plant fungi, and larvicidal activity against mosquitos and marine invertebrate larvae. Phlorotannins can reduce *P. aeruginosa* inflicted mortality in Caenorhabditis elegans.
4	Anti-tumor activity	[76,77]	Cytotoxicity; Apoptotic activity; Anti-proliferative; Reduces pro-apoptotic molecules; Carcinogenesis reduction.	*F. vesiculosus* samples exert specific cytotoxicity against tumor cell lines without affecting the viability of normal cells. Moreover, it was found that, among the nine different phlorotannin fractions tested, F5 (in presence of *eckstolonol* and phlorofucofuroeckol-A) was the most active against both Caco-2 colorectal and MKN-28 gastric cancer cells, inducing death via activation of both apoptosis and necrosis. The brown algae extract inhibited the alterations of F-actin arrangement and the downregulation of E-cadherin expression.
5	Antiviral activity	[78]	Growth inhibition	Among the twenty phlorotannins studied, eckol hexacetate, phlorofucofuroeckol, fucofuroeckol, and bifuhalol-hexacetate showed significant binding affinities across the selected targets. Phlorotannins could be therapeutic candidates against SARS-CoV-2.
6	Anti-allergic effects	[79,80]	Via inhibition of FcεRI expression; Inhibition of calcium influx; Inhibition of mast cell degranulation.	Extracts were able to act upon cellular events triggered by immunological reaction (IgE/antigen), and on cellular events downstream the Ca^2+^ influx caused by a chemical stimulus (calcium ionophore A23187), preventing degranulation of RBL-2H3 cells. Furthermore, a dose-dependent behavior towards allergy-related enzymatic systems was observed for all the phlorotannin extracts. The three phlorotannins isolated from *S. carpophyllum*, 2-[2-(3,5-dihydroxyphenoxy)-3,5-dihydroxyphenoxy]-1,3,5-benzenetriol (1), 2,2’-[[2-(3,5-dihydroxyphenoxy)-5-hydroxy-1,3-phenylene]bis(oxy)]bis(1,3,5-benzenetriol) (2), and 2-[2-[4-[2-(3,5-dihydroxyphenoxy)-3,5-dihydroxyphenoxy]-3,5-dihydroxyphenoxy]-3,5-dihydroxyphenoxy]-1,3,5-benzenetriol (3), which have a simple phloroglucinol polymer structure, inhibit β-hexosaminidase, PGD2, and TNF-α secretion from DNP-HSA-stimulated RBL-2H3 cells. These compounds likely regulate phosphorylation in the IκB-proteasome 20S pathway. Fuhalol-type phlorotannins appear to exhibit multiple antiallergic effects, such as affecting proteasome, ROS production and secretion mechanisms.
7	Anti-Tyrosinase activity	[81,82]	Reduce the cellular melanin content and tyrosinase activity; Downregulate the expression of melanogenesis enzymes including tyrosinase, tyrosinase-related protein (TRP)-1, and TRP-2 in B16F10 melanoma cells.	974-A was demonstrated for the first time to be a potent competitive inhibitor of mushroom tyrosinase activity towards l-tyrosine and l-DOPA (IC_50_ values = 1.57 ± 0.08 and 3.56 ± 0.22 µM, respectively). Compounds isolated from the marine seaweed *E. stolonifera* (974-A, phlorofucofuroeckol-A, and eckol) could be used as tyrosinase inhibitors and be further explored in the cosmetic and agricultural fields. Molecular structures of phlorotannins strongly affect their tyrosinase inhibitory activity.
8	Anti-diabetic activity	[83,84]	Decrease serum glucose level, serum total cholesterol, total triglyceride, liver malondialdehyde, and activities of both of α-amylase and glucosidase; Increase serum insulin, hepatic glutathione, and total antioxidant capacity; Reduced damage in β cells of pancreases.	Phlorotannins that isolated from *Cystoseira compressa* and applied to the diabetic rats with a dose of 60 mg/kg of phlorotannin extract caused a significant boost in catalase activity. The phenolic extract exhibited a half maximal inhibitory concentration (IC_50_) of α-amylase (47.2 ± 2.9 μg) and α-glucosidase (28.8 ± 2.3 μg) inhibitory activities.
9	Angiotensin I-converting enzyme (ACE) inhibitory activity	[85]	Combine with the ACE molecule but not with the active site. Form covalent bonds with proteins; The protein precipitation ability of phlorotannins varies in a pH-dependent and concentration-dependent manner.	Dieckol was the potent ACE inhibitor and was found to be a non-competitive inhibitor against ACE according to Lineweaver-Burk plots. Dieckol had an inducible effect on the production of NO in EAhy926 cells without having cytotoxic effect.
10	Hepatoprotective effects	[86,87]	Exhibiting the protective effect on liver cells injured by tert-butyl hyperoxide(t-BHP). The dieckol-rich extract could prevent hepatotoxic of ethanol, decrease content of the malondialdehyde (MDA) and efficacy on the antioxidant defense system in mice.	The liver injury induced by intake of ethanol is associated with oxidative stress. Our results indicated that dieckol-rich phlorotannins (DRP) could reduce the ethanol induced liver injury in vivo through reducing the total cholesterol, inhibition of reactive oxygen species (ROS) generation and reduction of MDA formation. This hepatoprotective effect should due to the presence of bioactive compounds in the DRP. In addition, this study indicated that intake of DRP could be beneficial to the human health.
11	Neuroprotective effects	[2,88]	Multitarget ligands promoting; Modulate the activity of CNS enzymes and neuronal receptors; Regulating signaling pathways linked to oxidative stress-mediated neuronal cell death and neuroinflammation; Ameliorate the Aβ formation by modulating α-and γ-secretase expression and inhibiting Aβ-induced neurotoxicity.	Three major phlorotannins of *E. cava*-eckol, dieckol, and 8, 8‘-bieckol exhibited anti-apoptotic and anti-neuroinflammatory properties against Aβ-induced cellular damage.
12	Prebiotic effect	[89]	Enhance the levels of propionate and butyrate, which are two important short-chain fatty acids known for their role in intestinal homeostasis.	Phlorotannins from *F. vesiculosus* can positively contribute to the maintenance of a healthy gastrointestinal condition.
13	Anti- adipogenesis activity	[90,91]	Inhibit adipocyte differentiation and lipid formation/accumulation in 3T3-L1 fibroblasts.	Phlorotannins inhibited adipocyte differentiation by suppressing peroxisome proliferator activated receptor γ (PPARγ) and CCAAT/enhancer-binding proteins (C/EBPs) expression. These phlorotannins are promising candidates for the management of obesity. Anti-adipogenesis effect of phlorotannins at the concentration of 20 µM was observed by reduced lipid accumulation and the suppressed expression of lipogenic differentiation markers.
14	Laxative effects	[92]	Phlorotannins (Pt) treatment induces the recovery of stool parameters, GI transit, histopathological and cytological alterations, GI hormone concentrations and the mAChR signaling pathway in SD rats with loperamide (Lop)-induced constipation.	The laxative effects of Pt are associated with alterations of the fecal microbiota profile of SD rats with Lop-induced constipation. Pt compounds derived from E. cava are potential therapeutic candidates for the treatment of constipation.
15	Sedative–hypnotics	[93]	Promote non-rapid eye movement sleep in mice via the benzodiazepine (BZD) site of the GABA_A_ receptor; Produce a significant decrease in sleep latency and an increase in the amount of non-rapid eye movement sleep (NREMS).	The major phlorotannin constituent eckstolonol showed sleep-promoting effects via the BZD site of the GABA_A_ receptor. Phlorotannin preparation (PRT) decreased the mean duration of wake episodes and increased the total number of wake and NREMS bouts. These results clearly indicate that PRT inhibited the maintenance of wake.
16	Potential natural muscle building supplements	[94]	Downregulating the Smad-signaling, a negative regulator; Upregulating the insulin-like growth factor-1 (IGF-1) signaling, a positive regulator.	Of the six phlorotannin isolates evaluated, dieckol (DK) and 2,7”-phloroglucinol-6,6′-bieckol (PHB) induced the highest degree of C2C12 myoblast proliferation. DK and PHB bind strongly to myostatin, which is an inhibitor of myoblast proliferation, while also binding to IGF-1 receptors. DK and PHB are potential natural muscle building supplements and could be a safer alternative to synthetic drugs.

**Table 8 marinedrugs-20-00742-t008:** Advantages and disadvantages of substances derived from marine organisms.

Name	Advantages	Disadvantages
Chitosan	Non-toxic, biocompatible, biodegradable, non-allergenic. Parenteral and mucosal administrations. Controlled antigen release. Mucosal administration elicits robust antibody and T-cell responses.	Poor reproducibility of the results due to the variability of the chemical structure. Poor solubility above pH 6.
Fucoidans	Almost complete absence of toxicity, safety and excellent biocompatibility. Regulation of cellular and humoral immunity as well as hematopoietic mobilization. Potentiation of the function of immune cells. Anti-cancer effect.	Difficulties with obtaining structurally characterized and homogeneous samples or oligomeric fractions.
Carrageenans	No adverse side effects at intranasal use. An activation of macrophages. Induction of the generation of pro-inflammatory cytokines. Significant ability to enhance antigen specific immune responses as well as antitumor effects.	Limited solubility. Anticoagulant properties. Prolonged oral administration can lead to the development of inflammation of the gastrointestinal tract.
Alginate	Non-toxic, biocompatible, biodegradable. Mucoadhesive nature and a relatively low cost. Stimulation of Th1 response and production of specific antibodies. Anti-cancer and anti-allergic properties. An ability to form hydrogel microspheres and nanospheres which possess higher immunostimulatory effect.	Variable chemical structures.
MF59^®^ (Composed of squalene)	Compared to aluminum salts, MF59^®^ causes stronger immune response, stimulating both antibody production and T-cell immune response.	Reactogenicity. Pain at injection site. Induces inflammatory arthritis
AS03 (Composed of the same percentage of squalene and DL-α-tocopherol)	The strong stimulation of both antibody production and Th1 and Th2 immune response.	An association between the AS03-adjuvanted Pandemrix vaccine and narcolepsy cannot yet be excluded

**Table 9 marinedrugs-20-00742-t009:** Studies of safety and toxicity of phlorotannins extracted from brown seaweeds, as measured in humans.

Composition of Participants	Extract	Raw Material	The Amount of Administration Given	Effect	Toxic Side Effects	Reference
Twenty-three participants (11 men, and 12 women) aged 19–59 years	Crude extract	*Ascophyllum nodosum* and *Fucus vesiculosus*	Oral, 1 capsule/day, treatment during 1 week, (250 mg/capsule)	Reduces insulin levels	No side effect	[103]
Twenty-four participants	*Dieckol*	*Ecklonia cava*	Oral, 2 capsules/day, treatment during 1 week (500 mg/capsule)	Promotes sleep	No serious adverse effects	[104]
Eighty participants aged 30–65 years	Crude extract	*Ascophyllum nodosum*	Oral, 1 capsule/day, treatment during 8 weeks (100 mg/capsule)	Reduces DNA damage	No side effect	[105]
107 participants (138 men, and 69 women) aged 19–55 years	Crude extract	*Ecklonia cava*	Oral, 1 capsule/day, treatment during 12 weeks (72 mg/capsule and 144 mg/capsule)	Anti-obesity potential	No side effect	[106]

**Table 10 marinedrugs-20-00742-t010:** Research progress of phlorotannins in food preservatives.

Brown Seaweeds	Excipients	Formulation	Fresh-Keeping Effect	Reference
Phlorotannins from *Sargassum tenerimum*	-	5% phlorotannins	A 4-days’ increase in the shelf-life of shrimp.	[151]
Phlorotannins (purity is 90%) extracted from kelp	Chitin (NCh, degree of acetylation: 90%)	1.5 g/kg NCh-phlorotannins	Sea bass fillets had lower bacterial growth, pH, TVB-N and TBA as well as better characteristics of texture, color and WHC than those of the control group during refrigerated storage.	[152]
Phlorotannin (Ph, puchased)	Sodium alginate (SA) poly(ethyleneoxide) (PEO) blended nanofibers	50:50:10 (SA/PEO/Ph)	Significantly increased the shelf life of chicken without altering their sensory quality.	[149]
Phlorotannin	*Momordica charantia* polysaccharide (MCP)	Phlorotannin (PT) was encapsulated in MCP nanofibers	After cold plasma treatment, the release efficiency of PT from the nanofibers was enhanced by 23.5% (4 °C) and 25% (25 °C), respectively. Correspondingly, both antibacterial and anti-oxidant activities of PT/MCP nanofibers were markedly improved.	[153]

## Data Availability

Not applicable.

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
