# Peer review of "A Bioactive Substance Derived from Brown Seaweeds: Phlorotannins"

_marinedrugs, 2022, doi:10.3390/md20120742_

Round 1

Reviewer 1 Report

I have read the manuscript and I have several questions and suggestions.
1. The authors did not state the purpose of their manuscript.
2. It is necessary to submit the section "Materials and Methods", in which please indicate the keywords, years of information search, criteria for inclusion / exclusion of literary references.
3. A large number of reviews over the past 5 years have been devoted to polyphenols (for example, https://doi.org/10.1016/j.fbio.2020.100832; https://doi.org/10.1016/j.algal.2021.102484; https:/ /doi.org/10.1016/j.foodres.2020.109589 and many others). How is your review different from others published?
4. Reference 27 does not contain information about the extraction of Ecklonia radiata and Carpophyllum plumosum. Please remove this confusion and give correct references.
5. The use of NADES as extraction solvents has recently attracted a large number of researchers. Review this section in more detail, please. Discuss the viability of this approach.
6. For the section "Application in medicine", it is necessary to provide data from clinical trials indicating the number, number of patients, type of study, reference drugs, doses, terms, results and conclusion.
7. Pharmacokinetics data are important for formulation development. The pharmacokinetics of marine molecules is extremely important (https://doi.org/10.3390/md18110557). Please include an additional section in section 5.1. Please discuss the relationship between lipophilicity/solubility and their bioavailability.
8. Phlorotannins can be used as a natural UV filter in sunscreen formulations. (for example, https://doi.org/10.1007/s10811-022-02705-2 etc.). Please review this aspect of the use of phlorotannins in Section 5.3.

Author Response

Please see the attachment:responses1.

Reviewer 2 Report

1.       Figure2. Please label the figure itself for easy follow-up.

2.       Please include a separate section mentioning the limitations of marine compounds and their clinical development as well.

3.       It would be nice if the authors include a table showing the pros and cons of marine compounds, especially described in the present manuscript.

4.       Authors have the opportunity to include some nice illustrations and for a review article that is a must to a clear message to the readers. Please include figures about mechanisms of action and the application of these products.

5.       Are there some compounds which already in clinical trials? Please induce the details, possibly in the form of a table. 

Author Response

Please see the attachment: Responses 2.

Round 2

Reviewer 1 Report

I have read the revised manuscript. The section "Materials and Methods" must be placed before the section "Conclusion"